# MODEL PRUNING WITH MODEL TRANSFER

## ABSTRACT

With a well-trained "full-size" network, model pruning aims to derive a small network by removing some weights with minimum performance deterioration, e.g., image classification accuracy. In the typical setup, both model training and pruning are done on the *same* target dataset, which represent a downstream task of interest. On the other hand, to better solve a downstream task in the real world, the well-established practice is transferring a model pretrained on some source data to the target dataset via finetuning. The two worlds motivate us to study model pruning in a new realistic setup, which embraces a pretrained model and allows transferring it to the target dataset. In the new setup, we first show, as expected, transferring a pretrained model improves state-of-the-art (SOTA) pruning methods remarkably once they follow a principled pruning pipeline: *transfer the pretrained model by finetuning on the target dataset, prune, and finetune again.* Surprisingly, in the new setup, the simplistic random pruning (which removes random filters) and the L1-norm method (which removes filters that have small L1 norms) outperform SOTA methods, and the latter performs the best! Based on the simple L1-norm method, we propose two techniques to further improve the pruning performance by exploiting the full-size model. Specifically, when finetuning the L1-norm pruned model, our techniques (1) directly reuse the full-size model's classifier, or (2) regularize the pruned model in its finetuning through aligning its features to the off-the-shelf class-mean computed by the full-size model. Extensive experiments on large-scale benchmark datasets demonstrate that our techniques significantly outperform existing approaches.

## 1 INTRODUCTION

Model pruning is an effective strategy to compress a neural network without notable performance deterioration by removing unimportant or redundant weights or filters. Pruning weights without considering structures is known as unstructured pruning (Han et al., 2015a;b; Hassibi & Stork, 1992; LeCun et al., 1989), which is straightforward to operate but requires dedicated sparse computation hardware or library for realistic acceleration (Han et al., 2016; 2017). In contrast, structured pruning removes part of the network structures such as filters instead of individual weights (Li et al., 2017; He et al., 2017; Luo et al., 2017; Wen et al., 2016). Structured pruning achieves inference speedup and run-time memory saving without demanding special software libraries or hardware designs (Li et al., 2017; Wen et al., 2016; He et al., 2017; Luo et al., 2017).

**Motivation.** The contemporary setup of pruning explores methods how to train a full-size model, prune it, and finetune the pruned models on the *same* target dataset (Han et al., 2015b; Lin et al., 2020; Fang et al., 2023). This setup impoverishes the applicability of pruning methods, because, on the other hand, real-world applications find that transferring a pre-trained model (trained on some large-scale source data) to the downstream task achieves significantly better performance (Donahue et al., 2014; Sun et al., 2017; Radford et al., 2021). We are motivated to study pruning in a new realistic setup that embraces a pre-trained model (Fig. 1). The new setup enjoys both worlds: model transfer provides better features or a pre-trained model to better serve the downstream task, and model pruning compresses the final model which can be readily deployable in mobile devices. Aiming for readily deploying a pruned model, we focus on structured pruning in this work. It is important to note that, while pruning and transfer learning are extensively and independently studied in the literature, our setup of studying *model pruning with model transfer* (Fig. 1) is underexplored yet of practical significance.

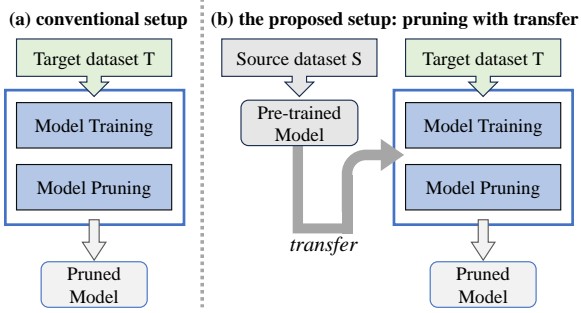

Figure 1: Compared to the conventional setup of pruning (Han et al., 2015a), the proposed setup embraces pretrained models (trained on some source data), allowing transferring pretrained models to better serve the downstream task. The new setup fits better to today's application scenarios and enriches the research of model pruning. The research focus, marked by blue box, opens up questions such as whether/when to transfer a pretrained model in pruning, how to finetune pruned models, how to improve pruning further, etc.

**Technical insights and salient results.** Numerous real-world applications show that transferring a (full-size) pretrained model significantly improves the performance on downstream tasks. We expect that doing so also improve the performance of pruned models, although typical pruning setup does not embrace pretrained models. Indeed, our experiments demonstrate that simply incorporating a pretrained model in pruning (Fig. 1b) significantly improves the performance of diverse approaches. We find a principled, somewhat intuitive, three-step pipeline for them to achieve improvements: *(1) transfer the full-size pretrained model by finetuning it on the target dataset, (2) prune the finetuned model, and (3) finetune the pruned model on the target set.* However, surprisingly, following this principled pipeline, random pruning, which removes random filters of the full-size model, rivals sophisticated pruning approaches; the simple $L_1$-norm method, which removes filters that have small norms, remarkably outperforms existing methods! Building on the $L_1$-norm method, we present simple and effective techniques to improve model pruning further. Our techniques are based on the observation that, the full-size finetuned model resoundingly outperforms its pruned versions at high pruning rates, indicating that the former's classifier and features can be used to guide the finetuning of pruned models. Hence, we exploit the full-size finetuned model (cf. model after the first step) to help finetune the pruned model (i.e., at the third step). Concretely, we propose to (a) reuse the full-size model's classifier, or (b) regularize the pruned model in its finetuning through aligning its features to the off-the-shelf class-mean computed by the full-size model. Extensive experiments show that our methods significantly outperform existing ones at a wide range of pruning rates, achieving the state-of-the-art on large-scale benchmark datasets.

**Contributions.** We make three major contributions.

1. We formulate a new, realistic setup of pruning by embracing pretraining models. This setup not only yields much better performant pruned models but also enriches the exploration space, e.g., whether to prune the pre-trained model, perform pruning before or after transferring the model on the target dataset, etc.

2. In the new setup, we extensively study existing pruning methods and find them to perform significantly better once following a principled pipeline: *transfer the pretrained model by finetuning on the target dataset, prune, and finetune again.* Surprisingly, we find that the simple L1 method, which prunes filters that have small norms, outperforms sophisticated approaches that are developed in the conventional setup (Fig. 1a).

3. We further propose simple techniques that reuse the full-size model's classifier and/or per-class mean features as auxiliary supervision to help finetune pruned models. Extensive experiments show that our techniques significantly outperform existing methods on diverse benchmark datasets.

## 2 RELATED WORK

**Model pruning** is an important topic in the area of model compression. Pruning methods can be categorized into two types: unstructured pruning, and structured pruning (Anwar & Sung, 2016). Unstructured pruning produces irregular sparse models, which require dedicated computing hardware or software library for real-world acceleration (Han et al., 2016; 2017). In contrast, structured pruning yields regular sparse models, readily deployable with acceleration on general-purpose hardware. Structured pruning can remove either individual filters or a full layer of parameters. The latter considers computation latency, yields a shallow network that loses some "deep capacity" and cannot

guarantee the performance of the pruned models (Wen et al., 2016; Jordao et al., 2020; Chen & Zhao, 2018). In contrast, filter pruning keeps deep network architectures, offers flexibility in structured pruning patterns, and is a more prevalent type of structured pruning. We explore filter pruning methods under the new setup (Fig. 1b) which embraces a pretrained model.

There are plenty of filter pruning methods (Liu et al., 2018; Li et al., 2017). To name a few, L1 method (Li et al., 2017) computes the L1-norm of each filter and removes those that have small norms; Network Slimming (Liu et al., 2017) imposes an L1 regularizer on the scaling factors in Batch Normalization layers during training and removes filters that have small scaling factors; SFP (He et al., 2018) alternates pruning and training towards learning a pruned model; PFS (Wang et al., 2020) removes filters from randomly initialized full-size models and trains the pruned model from scratch; HRank (Lin et al., 2020) removes filters which produce low-rank feature maps. It is worth noting that DepGraph (Fang et al., 2023), a recently published method, achieves the state-of-the-art structured pruning performance and is able to prune any network architectures by explicitly modeling the dependency between layers and comprehensively grouping coupled parameters. We extensively compare against the prior arts in the new setup, and demonstrate that our simpler methods resoundingly outperform them.

**Model pretraining and transferring.** Numerous applications benefit from the de facto practice of transferring pre-trained models (Sharif Razavian et al., 2014). Concretely, models are pretrained on external large-scale source data and then finetuned on the target dataset which represents the downstream task. Model pretraining per se is an important research topic, concerning how to effectively exploit large-scale data, including both labeled (Khosla et al., 2020) and unlabeled data (He et al., 2020; Chen et al., 2020), single-modal or multi-modal data (Radford et al., 2021; Jia et al., 2021), etc. When trained on unprecedented scale of data, pretained models serve as a foundation which can be readily used in some downstream tasks (Yuan et al., 2021; Bommasani et al., 2021; Fei et al., 2022). Importantly, transferring them through finetuning significantly improves the performance on downstream tasks (Yuan, 2023). While the conventional setup of pruning does not consider using pretrained models, we formulate a new, realistic setup to study pruning by embracing pretrained models, allowing us to develop practically useful pruning methods in the modern era.

**Model-supervised learning** means the practice of exploiting a (larger) pretrained model to help train (smaller) better models. Various fields find this strategy to be effective, such as knowledge distillation, self-training, domain adaptation, etc. Knowledge distillation uses the pretrained teacher model's features or classifiers to help train a smaller student model (Yang et al., 2020; Wang et al., 2021). Self-training runs a pretrained model on unlabeled data to derive pseudo labels, which in turn are used to train a better model (Fralick, 1967; Lee et al., 2013). Unsupervised domain adaptation repurposes the pretrained model to help adapt the model in a new domain by aligning the features between old and new domains of data (Ganin & Lempitsky, 2015). Inspired by these lines of work, we use the full-size model to improve pruning performance, in a way of using its features or classifiers to help finetune the pruned model on the target dataset. While the strategy of using pretrained models to help train better models finds success in various fields, to the best of our knowledge, our work makes the first attempt to use such to improve pruning.

## 3 MODEL PRUNING WITH MODEL TRANSFER

We present the new setup of Model Pruning with Model Transfer, including problem formulation, training and evaluation protocols, metrics, datasets, and interesting research questions.

### 3.1 PROBLEM SETUP AND PROTOCOLS

Compared to current setup of deep neural network pruning which studies methods of training and pruning on the same target dataset, our suggested setup further embraces pretrained networks (Fig. 1b). That said, the pretrained networks are trained on some large-scale source data and allows model transfer before pruning in downstream tasks. In this sense, this new setup is more realistic in the current standard given various of publicly available pretrained deep models.

**Training protocol.** The new setup embraces a pretrained model (trained on some source data) but does not require such source data to be available. This is realistic in real-world scenarios. For example, source data might be too large in scale (e.g., LAION-2B contains 2 billion of image-text

Table 1: **A summary of datasets used in our study.** We use two large-scale, well-established datasets ImageNet-21k and ImageNet-1k as the source datasets to pre-train deep neural networks. Because the two datasets are widely used in computer vision research, there are plenty of publicly released models of various architectures free of use for academic purpose. We use three distinct datasets to simulate downstream tasks, spanning fine-grained recognition, long-tailed recognition, and general object recognition.

| source dataset | #train-images | #val-images | #classes | task tags |
|---|---|---|---|---|
| ImageNet-21k (Deng et al., 2009) | 14,197,122 | - | 21,843 | general |
| ImageNet-1k (Russakovsky et al., 2015) | 1,281,167 | 50,000 | 1,000 | general |
| target dataset | #train-images | #val-images | #classes | task tags |
| CUB-200-2011 (Wah et al., 2011) | 5,994 | 5,794 | 200 | fine-grained |
| CIFAR-100 (Krizhevsky et al., 2009) | 50,000 | 10,000 | 100 | general |
| iNaturalist 2018 (Van Horn et al., 2018) | 437,513 | 24,426 | 8,142 | long-tail, fine-grain |

paris (Schuhmann et al., 2022)) or has data privacy and protection issues (Horvitz & Mulligan, 2015; Radford et al., 2021). Therefore, in the new setup, if using a pretrained model, the training stage means finetuning it on the target dataset. Otherwise, training is literally mean training a model from scratch on the target dataset and/or finetuning a pruned model on the same target dataset.

**Evaluation protocol.** We study pruning methods by letting them prune at various pruning rates, ranging from 30% to 90%. Following the literature of pruning, we repeat each pruning method five times at each pruning rate and report the mean performance with standard deviation. As our work studies pruning through the lens of image classification, we use the classification accuracy as the metric to gauge the performance of a pruning method.

**Interesting questions.** The new setup tempts us to answer the following questions in this paper.

- Does transferring a pretrained model improve the performance of existing pruning methods?
- Is there a principled pipeline for them to shine when embracing a pretrained model?
- Do better-benchmarked methods in the typical setup still perform better in the new setup?
- How can we improve pruning performance further in the new setup?

### 3.2 Benchmark Datasets

Below we present benchmark datasets used in our work. Importantly, we introduce source datasets used for pretraining only, and target datasets used for benchmarking model pruning. Table 1 summarizes the source and target datasets. We hope our standardization effort will foster the research of pruning in the community.

**Source dataset.** In this work, we consider datasets of labeled images, allowing typical supervised learning to pretrain a deep network through image classification. We use two well-established large-scale image datasets: ImageNet-1K (Russakovsky et al., 2015) and ImageNet-21K (Deng et al., 2009), which contain 1.28M and 14M images, respectively. These datasets are widely used to study pretraining methods (Kolesnikov et al., 2020; Dosovitskiy et al., 2020; Liu et al., 2022b).

**Target dataset.** To evaluate pruning methods, we use three well-established datasets as target datasets, representing distinct downstream tasks. They are CUB-200-2011 (Wah et al., 2011), iNaturalist 2018 (Van Horn et al., 2018), and CIFAR-100 (Krizhevsky et al., 2009). CUB-200-2011 and iNaturalist are popular datasets for studying fine-grained visual categorization, and the latter is also a popular dataset to study long-tailed recognition consisting 8,142 classes. CIFAR-100 contains 100 classes of objects and is widely used in the literature of pruning. All the datasets are publicly available free of use for academic purposes.

## 4 Methodology: Baselines and Improved Approaches

We note that it is hard to revisit all the pruning methods in the new setup. Because numerous pruning methods exist that can be categorized into distinct types w.r.t different aspects and can be variants of one another. Moreover, most methods can work only with specific network architectures, e.g., convolutional neural networks or transformers. So we focus on some representative and most recent ones, which belong to structured pruning. We present them below, and introduce our novel techniques later in this section.

## 4.1 Baseline and Representative Methods

We present baseline and representative methods below. We describe the pruning pipelines in the next subsection, e.g., "training $\mapsto$ pruning $\mapsto$ finetuning" (Han et al., 2015a; Liu et al., 2018).

- **Random** (Mariet & Sra, 2016) randomly pruning filters or neurons regardless of its importance. It is perhaps the most naive pruning method. It has been treated as an uncompetitive baseline until recently that multiple works show its competitiveness with properly sparse-training or finetuning (Mittal et al., 2018; Li et al., 2022; Liu et al., 2022a).

- $L_1$-**norm** pruning method first computes the $L_1$ norm of each filter at each layer, removes a fixed percentage of filters that have the smallest norms (Li et al., 2017).

- **HRank** (Lin et al., 2020) removes filters that produce low-rank feature maps. Concretely, given a batch of training examples, for each filter, it performs Singular Value Decomposition on its resultant feature maps, records the ranks, and removes a predefined percentage of filters that correspond to the lowest ranks at each proper layer.

- **Network Slimming** (Liu et al., 2017) imposes an $L_1$ sparsity on channel-wise scaling factors in Batch Normalization layers during training (or finetuning the pretrained model), removes a predefined percentage of filters that correspond to the lowest scaling factors.

- **EPruner** (Lin et al., 2021) uses the Affinity Propagation method (Frey & Dueck, 2007) to cluster filters from all layers, keep the representative filters and remove those that can be represented or redundant.

- **DepGraph** (Fang et al., 2023) is the first structured pruning method that applies to various network architectures including CNN, Transformer, RNN, and GNN. It builds a dependency graph between filters across layers, modeling the structural coupling pattern to simultaneously remove structured parameters. DepGraph achieves the state-of-the-art performance on diverse benchmark datasets.

Most pruning methods are developed to prune filters in CNNs until recently that a few methods are proposed to prune Transformer networks. While **DepGraph** is the most recent and the state-of-the-art method that can prune both CNNs and Transformers, the two rather simple baselines **Random** and $L_1$-**norm** can also be used to prune CNNs and Transformers.

## 4.2 Pruning Pipelines

In the typical pruning setup, it widely recognizes the three-step pruning pipeline: "training $\mapsto$ pruning $\mapsto$ finetuning" (Han et al., 2015a; Liu et al., 2018). In contrast, adopting a pretrained model enriches the exploration of the pruning pipeline by allowing extra steps within the existing pipeline. Below we describe pruning pipelines with notations.

- **S**$\mapsto$**T** means that we pretrain a full-size model on the **S**ource dataset, transfers it to the **T**arget dataset through finetuning. This is a conventional pipeline of model transfer, typically used as a reference or upper bound to analyze the decreased performance of pruned models.

- **SP**$\mapsto$**T** pretrains a full-size model on the **S**ource dataset, **P**runes it, and transfers the pruned model to the **T**arget dataset through finetuning. In particular, superscripts $\mathbf{P}^w$ and $\mathbf{P}^r$ mean that, when finetuning the pruned model after pruning, we inherit the **w**eights or **r**andomly initialize the weights (i.e., only using pruned network architecture (He et al., 2015)), respectively. Their comparisons help us understand the importance of using the pretrained weights versus the pruned architecture.

- **S**$\mapsto$**TP**$^w$$\mapsto$**T** pretrains a full-size model on the **S**ource data, transfers to the **T**arget data through finetuning, **P**runes it and inherits **w**eights (i.e., P$^w$), and finetunes on the **T**arget data again.

- **TP**$^w$$\mapsto$**T** train a full-size model on **T**arget dataset only, **P**runes the model and inherit weights (i.e., P$^w$), and then finetunes on the **T**arget dataset again. This pruning pipeline is suggested one in the conventional setup (Han et al., 2015a; Frankle & Carbin, 2018; Liu et al., 2018), i.e., without adopting a pretrained model.

### 4.3 IMPROVING PRUNING VIA FEATURE ALIGNMENT

**Motivations and insights.** At increasingly large pruning ratios, the accuracy of pruned models on the target dataset decreases rapidly, compared to the finetuned full-size model (i.e., $\mathbf{S} \mapsto \mathbf{T}$). While this is a rather common observation, we are motivated to exploit the full-size model to help finetune the pruned model towards better accuracy. To this end, we study two simple methods: (1) reusing the classifier of the full-size model for the pruned one, and (2) using the features computed by the full-size model to regularize those computed by the pruned model.

**Method-1: Classifier reuse** directly uses the classifier of the finetuned full-sized model $\mathbf{S} \mapsto \mathbf{T}$ as that of the pruned model. During finetuning the pruned model, the classifier is frozen without updating. This method is inspired by the literature of Knowledge Distillation (Yang et al., 2020; Chen et al., 2022), in which various methods use the classifier of the teacher model to help train a student model. By analogy, we treat the finetuned full-size model as the teacher, and the pruned model as the student. We also compare results by updating this classifier but find it to perform less well (Fig. A4).

**Method-2: Feature Regularization** uses features extracted by the full-size model to regularize the fineuning of the pruned model. It is inspired by multiple lines of work. For example, Knowledge Distillation methods often use deep features extracted by the teacher model to supervise the training of the student (Hinton et al., 2015; Chen et al., 2021); finetuning multimodal pretrained models demonstrates better results if using features of a modality-specific model to supervise the finetuning of the other (Zhai et al., 2022; Li et al., 2023). In this work, we regularize the finetuning of the pruned model using features extracted by the full-size model $\mathbf{S} \mapsto \mathbf{T}$. Concretely, we first extract features of training data with the full-size model, denoted as $\mathbf{f_i}$ for example-$i$. We then compute the mean feature, denoted by $\mathbf{c}_k = \frac{1}{|\mathcal{I}_k|} \sum_{j \in \mathcal{I}_k} \mathbf{f}_j$ for class-$k$, where $\mathcal{I}_k$ is the set of training data belonging to class-$k$. For the $i^{th}$ training sample which is from class-$y_i$, we use class-mean features of $\mathbf{S} \mapsto \mathbf{T}$ as extra supervision to regularize direction of features computed by the pruned model using a simple loss term:

$$\mathcal{L}_d = 1 - \cos(\mathbf{f}_i, \mathbf{c}_{y_i}) \tag{1}$$

We apply this loss at the penultimate layer together with the CE loss to finetune the pruned model, and study the role of **Method-1** and **Method-2** in Sec. 5.5.

## 5 EXPERIMENTS

We carry out experiments to study baseline and state-of-the-art methods under the new setup, answering the interesting questions listed in Sec. 3.1, and validate our methods. We start with experimental settings including implementations and model architectures.

### 5.1 EXPERIMENTAL SETTINGS

**Model architectures.** We study pruning methods with the popular convolutional neural network architecture and the Transformer architecture. Specifically, we use ResNet-50 (He et al., 2016) and ViT-B-16 (Dosovitskiy et al., 2020). We pre-train the former on the ImageNet-1K dataset (Russakovsky et al., 2015) and the latter on the ImageNet-21K dataset (Deng et al., 2009).

**Implementations.** We use the open-source PyTorch toolbox to train models in experiments. We follow the literature of pruning to set hyper-parameters. Following Liu et al. (2018), we adjust the training epoch to ensure the same number of total floating point operations (FLOPs) throughout the training process. For example, we train the $\mathbf{S} \mapsto \mathbf{T}$ models for 100 epochs on a target dataset; when pruning 50% of FLOPs, the training epoch for the pruned model is increased to $100/(1 - 50\%) = 200$. The learning rate decay is also scaled proportionally. We experiment with multiple pruning methods (see Sec. 4.1) within a wide range of pruning rates ranging from 30% to 90%.

For CUB-200-2011 and CIFAR-100, we train the $\mathbf{S} \mapsto \mathbf{T}$ models for 120 / 80 epochs, with batch size as 32, SGD optimizer with momentum of 0.9, weight decay of 5e-4. The initial learning rate is set to 1e-3 with the linear learning rate schedule. On iNaturalist2018, we follow (Alshammari et al., 2022; Kang et al., 2020) to train the model $\mathbf{S} \mapsto \mathbf{T}$ for 90 epochs with batch size 512, set weight decay as 1e-4, use SGD optimizer with momentum 0.9 and cosine learning rate schedule (Loshchilov & Hutter, 2016) which gradually decays learning rate from 0.02 to 0.

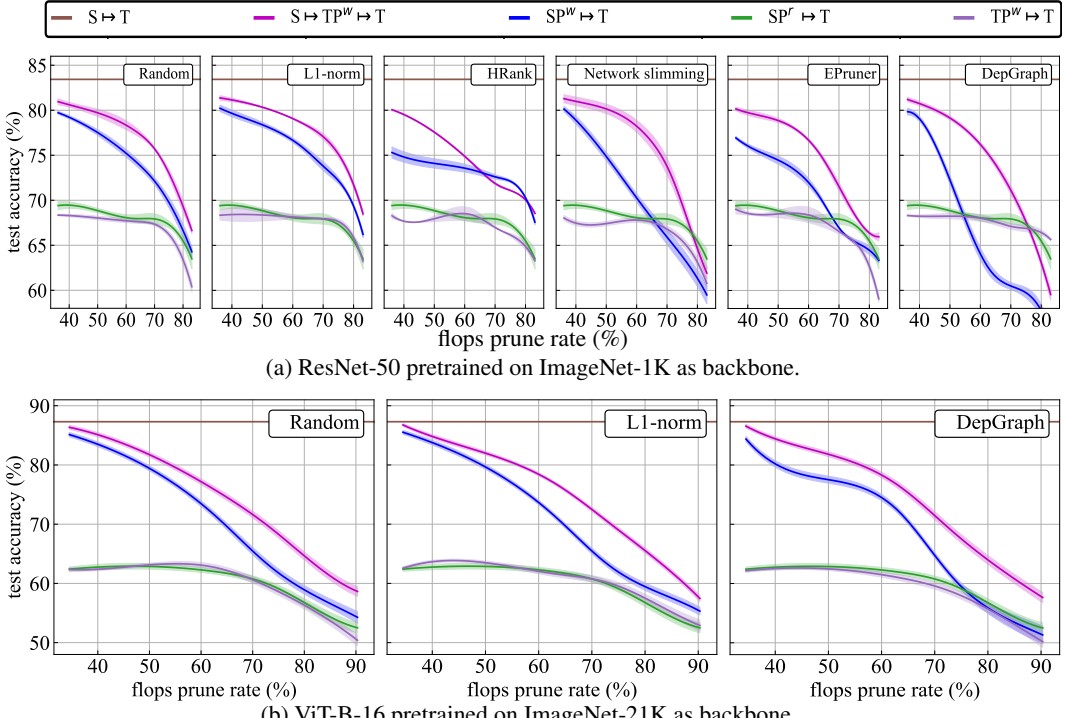

Figure 2: **Accuracy at different pruning rates** on the target dataset CUB-200-2011 by different pruning methods with different pretrained models (upper: ResNet-50 pretrained on ImageNet-1K, bottom: ViT-B-16 pretrained on ImageNet-21K). We display more results in Appendix B. Importantly, conclusions hold across datasets and network architectures of pretrained models. (1) Compared against methods in the typical setup which requires pruning and training on target dataset only ($TP^w \mapsto T$), transferring a pretrained model ($SP^w \mapsto T$) significantly improves their performance. (2) Crucially, transferring a model should (obviously) inherit the weights of the pretrained model; only transferring its architecture barely helps ($SP^r \mapsto T$). (3) Compared to the upper bound performance achieved by transfer learning without pruning ($S \mapsto T$), there is still a huge performance gap, implying fruitful research space in this new setup.

We adopt standard image augmentation techniques during training and finetuning including random-flip and rand-scale (He et al., 2016). Images are 224×224 in pixel resolution, randomly cropped from an augmented image. We apply mean subtraction and deviation division to preprocess images. In testing, we evaluate the center-crop view of the original image. We report the mean accuracy on the validation with standard deviation over five runs at each pruning rate for each method.

## 5.2 Message-1: Improving model pruning with model transfer

We investigate whether transferring a pretrained model (on a source dataset) improves the performance of pruned models. We benchmark various pruning methods on the CUB-200-2011 dataset. We compare against methods developed in the typical setup, i.e., $TP^w \mapsto T$, without adopting any pre-training. We test different pruning pipelines and configurations: inheriting pretrained weights ($SP^w \mapsto T$ and $S \mapsto TP^w \mapsto T$). For a complete comparison, we also report results of $SP^r \mapsto T$ which only inherits the network architecture but not the pretrained weights, and the finetuned full-size model $S \mapsto T$ which can be thought of as a performance upperbound. Fig. 2 lists the results.

With pruning rate in $[30\%, 60\%]$, both $SP^w \mapsto T$ and $S \mapsto TP^w \mapsto T$ significantly outperform $TP^w \mapsto T$, convincingly showing that pre-training can significantly improves the performance of pruned models. At increasingly higher pruning rates (e.g., $>65\%$), the margin between $SP^w \mapsto T$ and $TP^w \mapsto T$ decreases. For automatic pruning methods (Network slimming, EPruner and DepGraph), accuracy of $SP^w \mapsto T$ even becomes lower than $TP^w \mapsto T$, presumably because these methods may *automatically* remove an excessive proportion of filters in certain layers based on the global threshold, yielding sharp decreases in performance. As expected, only transferring architecture without the pretrained weights does not help pruning (cf. $SP^r \mapsto T$ vs. $SP^w \mapsto T$).

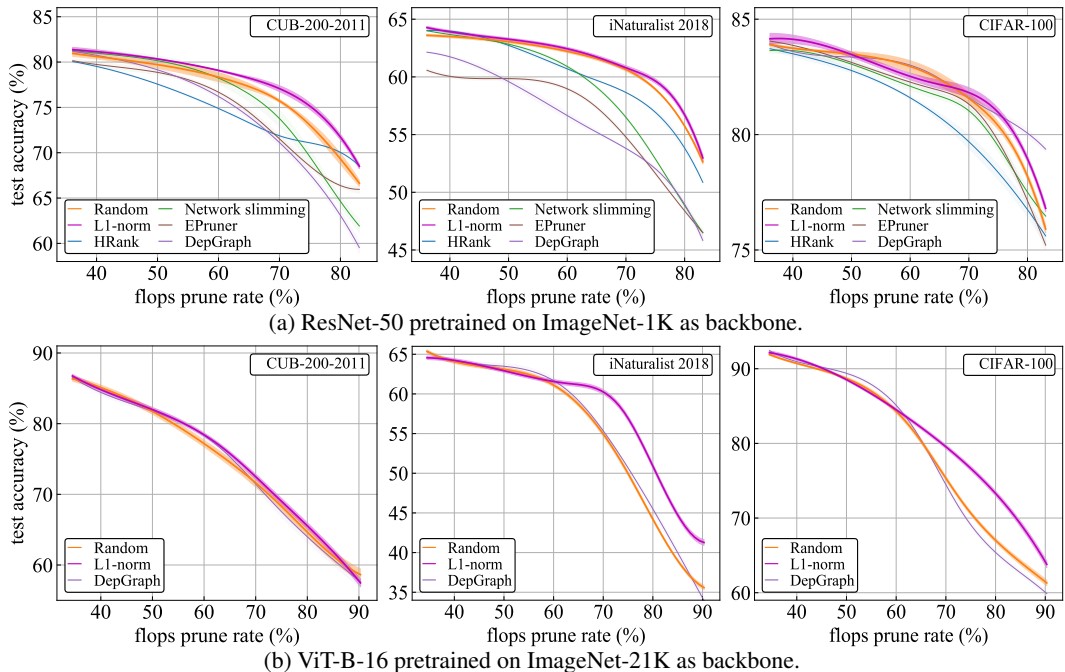

(a) ResNet-50 pretrained on ImageNet-1K as backbone.

(b) ViT-B-16 pretrained on ImageNet-21K as backbone.

Figure 3: **In the new setup, $L_1$-norm pruning method performs the best among all the compared approaches**, including the state-of-the-art such as DepGraph (Fang et al., 2023), HRank (Lin et al., 2020), and EPruner (Lin et al., 2021). Interestingly, the Random pruning method rivals them too. Owing to the simplicity of $L_1$-norm pruning, we explore how to improve pruning in this work.

## 5.3 MESSAGE-2: THE PRINCIPLED PRUNING PIPELINE IS TRANSFER → PRUNE → FINETUNE

Based on the previous experiment, we compare pruning pipelines that inherit pretrained weights: $\mathbf{SP}^w \mapsto \mathbf{T}$, and $\mathbf{S} \mapsto \mathbf{TP}^w \mapsto \mathbf{T}$. They differ in the order of model pruning and transfer. $\mathbf{SP}^w \mapsto \mathbf{T}$ first prunes the pre-trained model and then transfers the pruned one to the target dataset. In contrast, $\mathbf{S} \mapsto \mathbf{TP}^w \mapsto \mathbf{T}$ first transfers a pretrained model to the target dataset by finetuning and then performs pruning. Fig. 2 displays their comparisons (more results in Appendix B).

$\mathbf{S} \mapsto \mathbf{TP}^w \mapsto \mathbf{T}$ significantly outperforms $\mathbf{SP}^w \mapsto \mathbf{T}$, under a wide range of pruning rates [30%, 90%] and with different pretrained models (ResNet-50 and ViT-B). Recall that Sec 5.2 shows transferring a pretrained model already significantly improves existing pruning methods. Current results provide a guideline in the new realistic setup — given a model pretrained on a source dataset, one should transfer it to the target dataset via finetuning, perform pruning and finetuning on the target dataset.

## 5.4 MESSAGE-3: SIMPLE PRUNING METHODS OUTPERFORM SOPHISTICATED ONES

One may be curious about which pruning method performs the best under the new setup following the principled pipeline (concluded by the previous experiments). Hence, we compare different methods with the pruning pipeline $\mathbf{S} \mapsto \mathbf{TP}^w \mapsto \mathbf{T}$. Fig. 3 shows the comparisons. We observe that, surprisingly, both random pruning and the simple $L_1$-norm pruning method outperform sophisticated methods generally at all pruning rates, and the $L_1$-norm pruning performs the best! Results convincingly show that in the new setup which embraces a pretrained model, the simple $L_1$-norm pruning method shines and performs the best among various ones. Yet, it can be improved further as demonstrated next.

## 5.5 MESSAGE-4: IMPROVING PRUNING WITH FEATURE ALIGNING AND CLASSIFIER REUSE

We validate our proposed methods (Sec. 4.3) that reuse the classifier and/or per-class mean features of the full-size model (i.e., $\mathbf{S} \mapsto \mathbf{T}$) to regularize the finetuning of pruned models. We name them "w/ classifier" and "w/ class-mean", respectively. Based on previous results, we use the $L_1$-norm

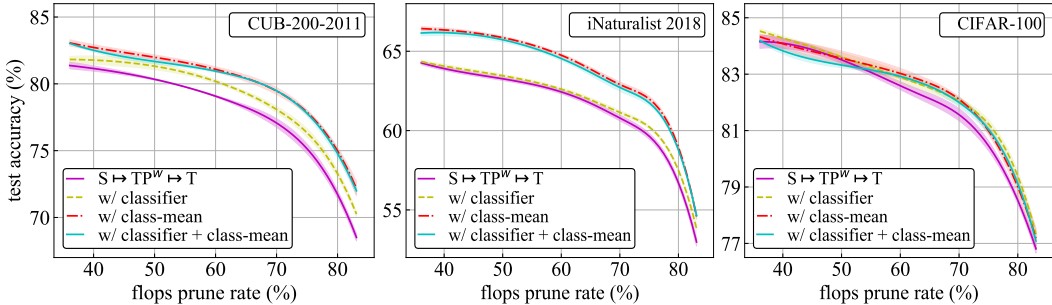

Figure 4: **Improving pruning using our methods**. The two proposd methods exploit the full-size model (after finetuning on the target dataset) to help finetune the pruned one. The first method "w/ classifier" reuses the classifier of the full-size model and freezes it during finetuning the pruned model. The second model "w/ class-mean" uses the class-mean features to regularize the features output by the pruned model being finetuned (Eq. 1). We analyze them with the best pruning pipeline (Fig. 2), by transferring the ImageNet-pretrained ResNet-50 model and atop of the $L_1$-norm pruning method which already yields the best performance (Fig. 3a), Results show that (1) our methods significantly improve over the $L_1$-norm pruning method, (2) simply using class-mean achieves the best performance gain, and (3) jointly using both does not necessarily improve further.

pruning method under the pipeline $\mathbf{S} \mapsto \mathbf{TP}^w \mapsto \mathbf{T}$ to evaluate our methods. Fig. 4 plots the results, showing that using either classifier or class-mean significantly improves the performance of pruned models, and the latter brings more remarkable performance gain! We also study the performance of applying both our methods; interestingly, "w/ classifier + class-mean" and "w/ class-mean" yield similar performance. We conjecture that regularizing features of the pruning model and learning a new classifier offers leeway to improve further, better than directly reusing previously finetuned classifiers of the full-size model.

## 5.6 Limitations and Societal Impacts

**Limitations.** We note several limitations of our work and hope future works will address them. First, while supervised learning is the mainstream strategy of pretraining and we only study pruning with supervised pretrained models, we believe it is worth using other pretraining models, e.g., self-supervised pretrained ones (Caron et al., 2021; Oquab et al., 2023) and language-supervised pretrained fundation models (Radford et al., 2021; Jia et al., 2021). Moreover, besides classification, we did not study pruning in other tasks such as detection and segmentation, which have wider range of real-world applications.

**Societal Impacts.** As our setup accepts pre-trained models, when inheriting pretrained weights, the resultant pruned models may inherit biases and unfairness learned by the full-size pretrained model. Moreover, it is unclear if pruned models are less robust than the pretrained full-size model. These are potential societal impacts and future work should analyze them.

## 6 Conclusion

We formulate a new, realistic setup to study pruning methods by allowing transferring a pretrained model to the target dataset. In the new setup, we revisit baseline and state-of-the-art (SOTA) pruning methods, which have been developed in the typical setup. As expected, they perform notably better with a pretrained model than without. We find that there is a principled pruning pipeline to make them shine: *transfer the pretrained model by finetuning on the target dataset, prune, and finetune again*. Yet surprisingly, in this setup, the simplistic $L_1$-norm method, which simply prunes filters that have small L1 norms, outperforms SOTA methods. Based on the simple L1 method, we propose two simple and effective methods to improve pruning. The methods exploit the full-size model to help finetune the pruned model: (1) directly using the full-size model's classifier as that of the pruned model, and (2) using class-mean features computed by the full-size model to regularize those from the pruned model being finetuned. Extensive experiments show that our approaches improve pruning significantly further.

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

# Appendix

## A SETUP AND PRUNE PIPELINE

We described the pruning pipelines under the setup of *pruning with transfer* in Sec. 4.2, and the flow chat of each pipeline is shown in Fig. A1. Different from the *typical* setup of pruning (i.e., $\mathbf{TP}^w \mapsto \mathbf{T}$), both training and pruning are performed on the same target dataset, we formulate a new and realistic pruning setup that allows transferring a pretrained model on large scale source dataset to the target dataset.

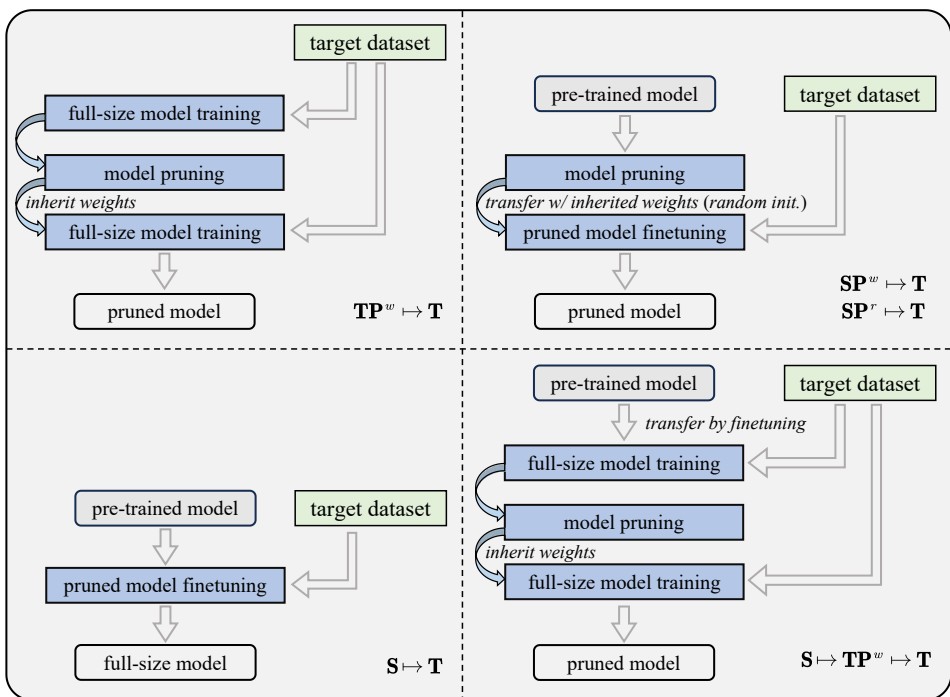

Figure A1: **The new setup enjoys both worlds**: pre-trained model to better serve the downstream task, and model pruning compresses the final model which can be readily deployable in mobile devices. **Upper left**: this pruning pipeline is suggested one in the conventional setup, i.e., without adopting a pretrained model. **Lower left**: this is a conventional pipeline of model transfer, typically used as a reference or upper bound to analyze the decreased performance of pruned model. **Right**: embraces a pre-trained model to better serve the downstream task, they differ in the order of model pruning and transfer.

## B MORE RESULTS ABOUT THE PRINCIPLED PRUNING PIPELINE

In Sec. 5.3, we concluded the principled pruning pipeline in the new setup, *prune with transfer*, that is, transfer→ prune→finetune on CUB-200-2011. Here, we also conduct experiments on iNaturalist 2018 and CIFAR-100, and compare the results of $\mathbf{SP}^w \mapsto \mathbf{T}$ and $\mathbf{S} \mapsto \mathbf{TP}^w \mapsto \mathbf{T}$ in Fig. A2 and Fig. A3, respectively. As can be seen, the conclusion is consistent with Sec. 5.3, that is, $\mathbf{S} \mapsto \mathbf{TP}^w \mapsto \mathbf{T}$ significantly outperforms $\mathbf{SP}^w \mapsto \mathbf{T}$, with different prune methods under a wide range of prunign rates, and with different pretrained models.

## C ANALYZE THE PERFORMANCE DIFFERENCES CAUSED BY THE ORDER OF PRUNING AND TRANSFER

We studied the impact of the relative order of network pruning and model transfer on target dataset in Sec. 5.3, and observed that the pipeline $\mathbf{S} \mapsto \mathbf{TP}^w \mapsto \mathbf{T}$ always significantly outperform than

$\mathbf{SP}^w \mapsto \mathbf{T}$. We analyze the differences between them in depth to reveal the main reasons for the performance gap. There are two differences between pipelines $\mathbf{S} \mapsto \mathbf{TP}^w \mapsto \mathbf{T}$ and $\mathbf{SP}^w \mapsto \mathbf{T}$. The one is the relative order of pruning and transfer, and the other is whether the weights of the *classifier* of the full-size model are inherited. Note that $\mathbf{S} \mapsto \mathbf{TP}^w \mapsto \mathbf{T}$ has inherited the weights of the classifier and fine-tuned all parameters (both backbone and classifier) on the target data. Therefore, we construct experiments that $\mathbf{SP}^w \mapsto \mathbf{T}$ also additionally inherits the weights of the classifier of the full-size model (dubbed "w/ classifier") and use [tuning] or [freeze] to indicate whether freezes the weights of classifier during the fine-tuning process on the target data.

Fig. A4 shows that by $\mathbf{SP}^w \mapsto \mathbf{T}$ w/ classifier produces similar results to $\mathbf{S} \mapsto \mathbf{TP}^w \mapsto \mathbf{T}$, regardless of whether the state of *classifier* is [freeze] or [tuning]. In other words, the main reason for the difference in accuracy between pipelines $\mathbf{SP}^w \mapsto \mathbf{T}$ and $\mathbf{S} \mapsto \mathbf{TP}^w \mapsto \mathbf{T}$ is whether the weights of the classifier are inherited. In addition, experimental results also show that freezing the weights of the classifier during fine-tuning can achieve higher accuracy.

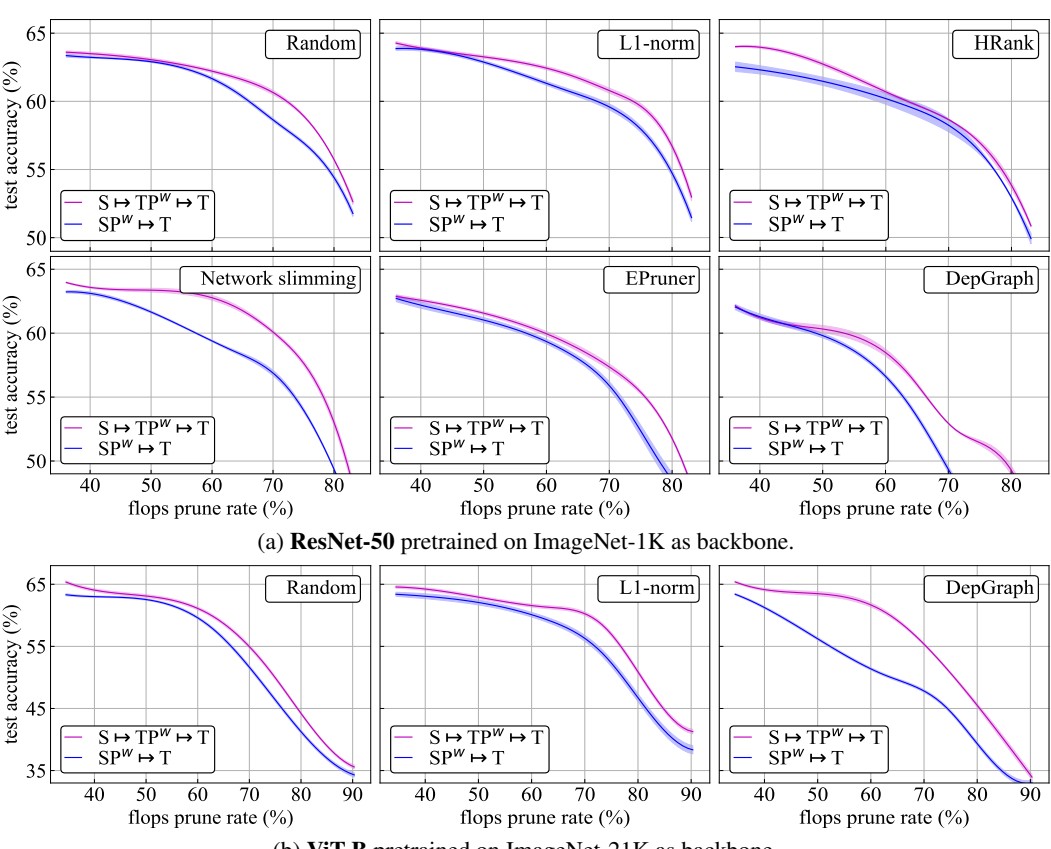

(a) **ResNet-50** pretrained on ImageNet-1K as backbone.

(b) **ViT-B** pretrained on ImageNet-21K as backbone.

Figure A2: Conclusion is consistent with Sec. 5.3, that is, $\mathbf{S} \mapsto \mathbf{TP}^w \mapsto \mathbf{T}$ significantly outperforms $\mathbf{SP}^w \mapsto \mathbf{T}$, with different prune methods under a wide range of prunign rates, and with different pretrained models. Here, the target dataset is **iNaturalist 2018**.

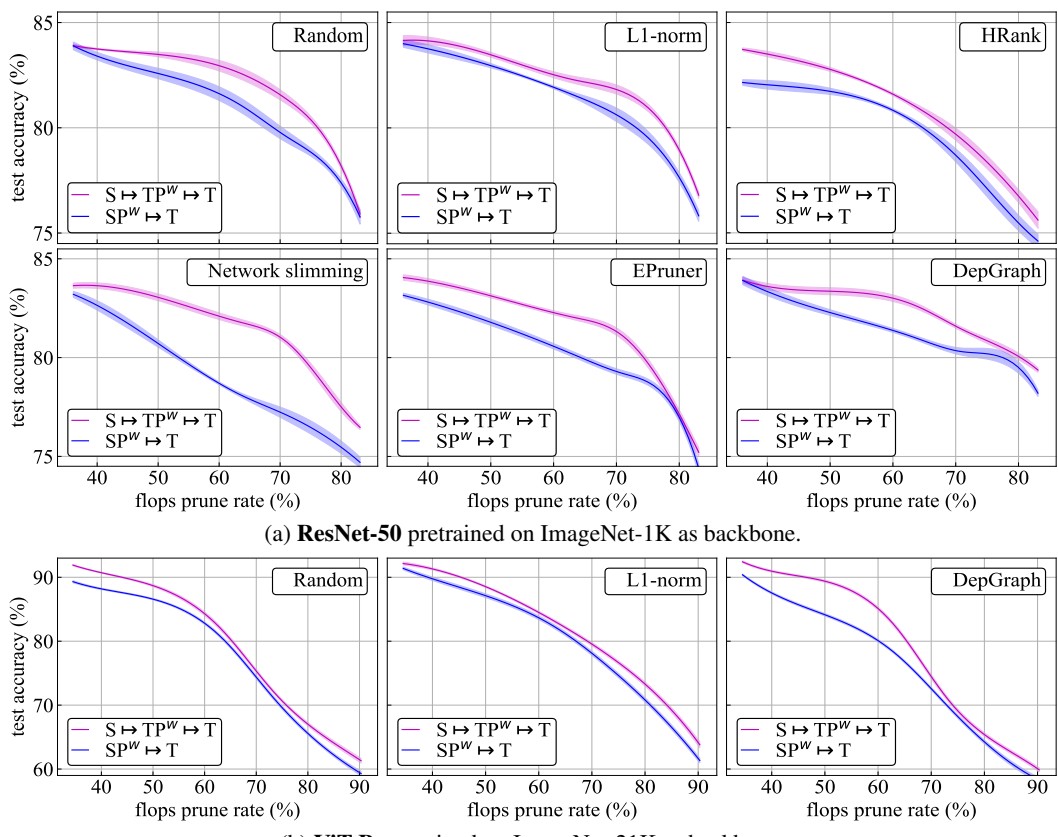

(a) **ResNet-50** pretrained on ImageNet-1K as backbone.

(b) **ViT-B** pretrained on ImageNet-21K as backbone.

Figure A3: Conclusion is also consistent with Sec. 5.3, that is, $\mathbf{S}{\mapsto}\mathbf{TP}^{w}\mapsto\mathbf{T}$ significantly outperforms $\mathbf{SP}^{w}\mapsto\mathbf{T}$. Here, the target dataset is **CIFAR-100**.

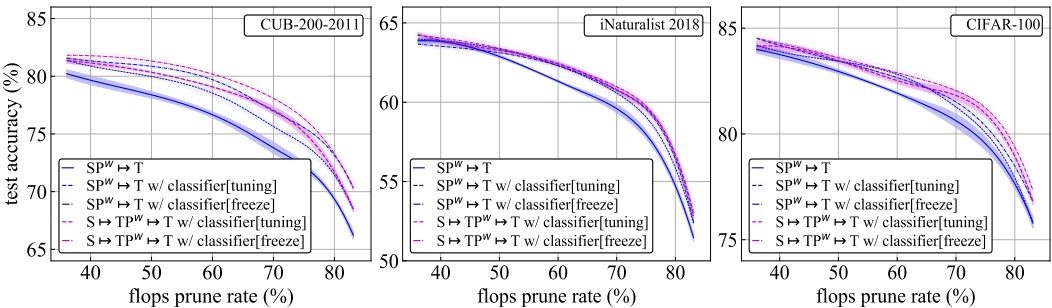

Figure A4: The performance difference between $\mathbf{SP}^{w}\mapsto\mathbf{T}$ and $\mathbf{S}{\mapsto}\mathbf{TP}^{w}\mapsto\mathbf{T}$ mainly lies in whether the weights of the classifier of the full-size model are inherited. During the fine-tuning process, freezing the classifier can achieve higher accuracy.

