# OpenReview forum: "Model Pruning with Model Transfer"
_ICLR.cc/2024/Conference — ICLR 2024 Conference Withdrawn Submission_

### Official Review · Reviewer_KtUB · 2023-10-30

**Soundness:** 3 good
**Presentation:** 3 good
**Contribution:** 2 fair
**Rating:** 5
**Confidence:** 3

**Summary:**

The papers explored multiple options in pruning the model in the transfer learning setting. Between finetuning and pruning at different stages of the process, the authors proposed an intuitive and principled way of finetuning both before and after the pruning step. This is corroborated with experiment results. in addition, the authors also demonstrated that using the original model as distillation could further improve the finetuning results on the pruned model.

**Strengths:**

- The paper is clearly written and carefully explained the different scenarios of finetuning/pruning that are compared in the experiments.
- The conclusion is clearly stated and supported by experiment results.
- The results from the paper has practical implications as all techniques mentioned are widely used in the community.

**Weaknesses:**

- While the paper is well-written with good details, I find it a bit too empirical and lacking significant contribution to the understanding of the underlying problem. It appears to be very intuitive that finetuning before pruning would better align the feature embedding and avoid potential difference in neuron importance in the L1 pruning process. I would like to see the author add more discussions on the observations, such as why L1 is able to beat other pruning methods with this additional finetuning step.

**Questions:**

- I wonder if the comparison between different settings are fair, since some have additional training steps than others. For example, $S\rightarrow TP^\omega\rightarrow T$ has an extra step of training over $SP^\omega\rightarrow T$. Would it be more reasonable to train equal amount of FLOPs in each case for the performance comparsion?

---

### Official Review · Reviewer_jc6u · 2023-10-30

**Soundness:** 2 fair
**Presentation:** 2 fair
**Contribution:** 2 fair
**Rating:** 5
**Confidence:** 4

**Summary:**

This paper proposes a new setting that combines model pruning and model transfer. The paper studies lots of possible methods and find a good simple pipeline: transfer the pretrained model by finetuning on the target dataset, then prune the network with L1, and finetune the networks. To further improve the performance, the paper proposes to re-use the full-size model to teach the pruned model in its funetuning. Experiments show significant improvement.

**Strengths:**

- The paper proposes to re-use the full-size model to teach the pruned model in its finetuning to improve the performance on model transfer and model pruning.

- Experiments show the effectiveness of the proposed method.

**Weaknesses:**

- In this paper, the main contribution is the point that re-uses the full-size model’s classifier and regularizes the pruned model in its finetuning through aligning its features. However, the basic insight is the knowledge distillation that uses a large model to teach a small model. Combining both model pruning and model distillation is not a new method considering existing methods, such as [r1,r2].

[r1] Combining Weight Pruning and Knowledge Distillation For CNN Compression, CVPR2021W

[r2] Prune Your Model Before Distill It, ECCV 2022

- It is suggested that the paper should review the works about model pruning+model distillation, and highlight the main differences and advantages when compared with these methods. Moreover, it is suggested to compare these methods in experiments.

**Questions:**

See [weakness]

---

### Official Review · Reviewer_Cwmj · 2023-10-31

**Soundness:** 2 fair
**Presentation:** 2 fair
**Contribution:** 1 poor
**Rating:** 5
**Confidence:** 5

**Summary:**

The paper presents a pruning method for task transfer, which, in my assessment, lacks the compelling and promising aspects it claims to possess. Some of the claims made seem to be somewhat overstated, which raises concerns about their validity. Given these issues, I am inclined to recommend rejection of the paper in its current form.

**Strengths:**

The authors have insightfully observed that within the paradigm of pretraining-transferring-pruning-finetuning, certain previous works may not perform as well as the baseline. This discovery is indeed valuable and contributes meaningfully to the field.

**Weaknesses:**

In reviewing the existing literature, it’s apparent that there have been several studies employing neural architecture adaptation/search methods to achieve impressive results on transferring tasks. In light of this, I would kindly suggest that the authors reconsider the scope of their claims and potentially draw comparisons between their proposed method and these existing lines of work.


The paradigm of pretraining-transferring-pruning-finetuning, as it is presented, seems to be very intuitive. It appears to lack novelty and does not seem to provide new insights.


The authors assert that their approach of incorporating pretraining on a large dataset for the transferring task represents a novel contribution within the pruning domain. While I recognize their perspective, I would like to offer some additional context from related fields for consideration. For instance, in the domain of language processing, practices such as pruning BERT models often involve pretraining on extensive datasets. Similarly, in object detection model pruning (e.g., with YOLO models) and semantic segmentation model pruning, large dataset pretraining is a common methodology. Given these examples, it might be worthwhile to reassess the uniqueness of the setup presented in this paper.


In this paper, the focus is primarily on transferring tasks related to classification. While classification is undoubtedly significant, I believe it would be beneficial to extend the investigation to include tasks such as object detection and semantic segmentation. These tasks are crucial in their own right and could provide a more comprehensive evaluation of the proposed method. I have some reservations about the effectiveness of the authors' method when applied to these additional tasks, and I think it would be valuable for the authors to address these areas to validate and possibly enhance the robustness of their method.



The technique employed in the paper is Knowledge Distillation (KD), which, in my opinion, is a fairly standard and not particularly novel approach. Its intuitiveness, while beneficial for understanding, could also be seen as a factor that diminishes its novelty.


The content in Section 5.2 seems to lack substantial information and depth, as it discusses the well-established fact that pretraining can enhance performance—a point already widely recognized in the field.


The message in Section 5.3 lacks sufficient information and is overly intuitive.

I have reservations about the content in Section 5.4, as I believe it may not consistently present an accurate picture. The primary distinction between this setup and a conventional pruning setup lies in its inclusion of a pretraining phase. However, I am of the opinion that this addition does not result in significant changes or improvements. On the other hand, I do recognize that more sophisticated pruning techniques, such as search-based pruning, can potentially yield better results and performance gains.


The utilization of Knowledge Distillation (KD) is widely recognized as a technique that can lead to performance improvements. Given this common knowledge, the content in Section 5.5 is informationless.

**Questions:**

See #Weakness.

---

### Official Review · Reviewer_LN4j · 2023-11-01

**Soundness:** 3 good
**Presentation:** 3 good
**Contribution:** 3 good
**Rating:** 3
**Confidence:** 4

**Summary:**

Motivated by compressing the neural network models with the pruning technique within same domain, this paper aims to extend it to different domain. To this end, this paper proposes the framework integrating the model pruning technique into the transfer learning problem; 1) finetunes the model in target domain, then 2) applies the pruning to the finetuned model, and 3) finetunes it again.
As a result, this paper found the proposed framework combining L1-norm outperforms SOTA methods.

**Strengths:**

This paper conducted various experiments to verify the proposed method and showed the sufficiently good performance.

**Weaknesses:**

As decribed in Sec. 5.6, this paper has applied the proposed method to the downstream tasks.
Albeit good motiviation, it seems to be a lack of review of the related works.
This reviewer suggests below papers that should be covered in this manuscript.
I suggest the authors clarify the difference and the relationship between this paper and [1, 2].

[1] Myung, S., Huh, I., Jang, W., Choe, J. M., Ryu, J., Kim, D., ... & Jeong, C. (2022, June). Pac-net: A model pruning approach to inductive transfer learning. In International Conference on Machine Learning (pp. 16240-16252). PMLR.

[2] Han, S., Pool, J., Narang, S., Mao, H., Gong, E., Tang, S., ... & Dally, W. J. (2016). Dsd: Dense-sparse-dense training for deep neural networks. arXiv preprint arXiv:1607.04381.

**Questions:**

1. I think the authors should cover and explain the papers proposed above. Actually, I know how the papers are different and related, but wouldn't readers be confused?
2. Sec. 5.1 explained that the authors adjusted the training epoch to meet the number of FLOPS. How does the result of Fig. 2 change if we train enough without such adjustment?